# Metal Peroxide Nanoparticles for Modulating the Tumor Microenvironment: Current Status and Recent Prospects

**DOI:** 10.3390/cancers16213581

**Published:** 2024-10-24

**Authors:** Jagadeesh Rajaram, Yaswanth Kuthati

**Affiliations:** 1Department of Biochemistry and Molecular Medicine, National Dong Hwa University, Hualien 974, Taiwan; jagadeesh31994@gmail.com; 2Department of Anesthesiology, Cathay General Hospital, Taipei 106, Taiwan

**Keywords:** metal peroxides, TME, cancer therapy

## Abstract

The International Agency for Research on Cancer (IARC) projects that by 2040, the global incidence of cancer would reach 27.5 million new cases. The current primary modalities of cancer treatment encompass surgery, radiation, and chemotherapy, which can harm normal tissues or fail to fully eliminate cancer. Nanomaterials make it easier to keep an eye on surgeries that remove tumors, deliver chemotherapy directly and selectively to cancer cells and neoplasms, and make radiation therapy more effective. It mitigates the risk and enhances the survival of cancer patients. This review aims to consolidate various metal peroxide nanomaterials utilized in cancer treatment with a special emphasis on metal peroxide nanoparticle surface modification for enhancing the efficacy of nanomaterials in tumor microenvironment. Researchers can acquire knowledge regarding metal peroxide nanoparticle types, their mechanisms of action, and their contributions to existing cancer medicines for enhanced cancer management.

## 1. Introduction

Cancer develops when there are irregularities in the DNA. These alterations cause uncontrolled cell growth, which in turn harms tissues [1]. Cancer is a disease characterized by a complex interplay between genetic and environmental factors, rendering it a multifactorial disease of origin [2]. Researchers identify DNA damage as the critical factor that plays a crucial role in the emergence of malformations, ultimately leading to the initiation of cancer. Cancer ranks among the top human diseases in terms of its impact on cause-specific disability-adjusted life years (DALYs), making it the most expensive disease. Estimates place the probability of developing cancer among individuals aged 0 to 74 years at 20.2%, with a slightly higher incidence of 22.4% among men and 18.2% among women [3]. The World Health Organization (WHO) has disclosed that cancer stands as the primary cause of mortality globally, resulting in around 10 million fatalities in the year 2020 [4]. As per the findings of Singhal et al. (2010), chemotherapy, radiation therapy, and surgical interventions constitute the three most prevalent methods of cancer treatment. Chemotherapy is a widely used approach to cancer management [5]. Conventional chemotherapy primarily targets cancer cells by disrupting mitosis and ceasing DNA replication [6]. Chemotherapeutic drugs unintentionally cause adverse and fatal side effects when they target healthy tissues such as the blood and digestive tract [7]. Moreover, radiotherapy presents a multitude of challenges because it directs radiation toward the healthy tissues near the tumor, potentially leading to unfavorable side effects [8]. The expense associated with surgical therapy is not only significant, but it also represents a highly intricate process that requires specialized knowledge. Furthermore, surgical therapy may require additional appointments for reasons such as suture removal, infection, contact dermatitis, suture expulsion, and graft examination. Moreover, a significant number of elderly patients require supplementary care post-surgery, including changes in dressings, transportation assistance, and subsequent visits, thereby escalating the cost of surgical therapy [9]. Additionally, the TME contributes to the recurrence of cancer in primary cancer patients. The TME plays a crucial role in regulating the aggressiveness, motility, dissemination, and colonization of cancer cells into distant organs. The various compositions of the TME include the extracellular matrix (ECM), the basement membrane (BM), endothelial cells, adipose cells, tumor-infiltrating immune cells, cancer-associated fibroblasts (CAFs), neuroendocrine cells, pericytes, and a plethora of signaling molecules that regulate tumor progression. Cancer cells secrete growth factors and cytokines (including IL-6, IL-1β, TGF-β1, TGF-β2, FGF-2, and PDGF), which help in the remodeling of the ECM and the development of the TME in cancer cells. Understanding the complex roles of TME constituents in various stages of metastatic progression will help study tumor development and also combat them [10].

Nanotechnology has paved the way for a new era in all areas of life. The real-time detection of molecular signals and biomarkers, facilitated by nanotechnology, has spurred considerable progress in the domains of early detection, diagnostics, prognostics, and therapeutic strategies. In the realm of cancer nanotherapeutics, nanotechnology has overcome several challenges, including nonspecific biodistribution, low water solubility, and insufficient bioavailability. It offers, in response, high sensitivity, specificity, and the capability for multiplexed measurements [11]. Nanomaterials have distinct advantages in cancer treatment, including biocompatibility, reduced toxicity, superior stability, improved permeability and retention impact, and precise targeting capabilities. Compared with traditional methods of drug delivery, nanoparticle-based systems demonstrate superior efficacy. This is because they can make substances that do not dissolve in water dissolve more easily, keep drugs and proteins working longer, and precisely control and direct the release of medicines at disease-affected body parts [12,13]. In addition, nanoparticles stand out with their high loading efficiency, stability, extended drug release, and capacity to deliver poorly soluble medications. The dynamic nature of nanotechnology can reduce limitations in chemotherapy and other therapies by developing various nanoparticles, such as protein-based nanoparticles, micelles, and polymer-based nanoparticles. Metal-based nanoparticles have gained prominence due to their unique physical properties and advantages in surface fictionalization. Targeting the TME through nanotechnology presents a significant challenge due to their complex structures and the various mechanisms involved in their development. Researchers developed various nanoparticles, such as polymers, liposomes, and nanogels, to target the TME’s pH environment, hypoxia, acidic environments, and oxidative stress Additionally, the incorporation of nanoparticles with various modalities such as chemotherapy, photothermal therapy (PTT), photodynamic therapy (PDT), radiotherapy (RT), and immunotherapy agents can achieve efficiency with zero negligence [14,15,16].

Metal nanoparticles (NPs) have garnered particular interest due to their potential as versatile therapeutic agents. These nanoparticles, constructed from a combination of gold, silver, iron and/or iron oxide, zinc, titanium, cerium oxide, nickel, copper, magnesium, barium, calcium, and bismuth, have been subject to thorough examination for their potential in cancer therapy. Metal nanoparticles play a pivotal role in contemporary cancer research, and their growing interest is evident. Gold nanoparticles are the most promising, followed by silver nanoparticles and magnetic nanoparticles (MNPs) when treating various diseases. Numerous studies have demonstrated the potential of metal NPs in cancer therapy, with both preliminary and clinical trials currently in progress [17,18]. In addition, metal nanoparticles have properties such as avoiding sudden drug release and off-target effects, which provide many advantages such as improved pharmacokinetic properties and the flexible control of drug release, helping them to target the TME with more precision. Alterations of MNPs with exterior molecules or inherent characteristics can adapt to the distinct chemical attributes of the TME by accurately delivering immune-modulating components to designated tissues or particular immune cells, securing a potent and improved immune reaction for better therapeutic results [19,20,21]. The utilization of non-noble metal-based therapies for cancer treatment could lead to more cost-effective alternatives to the expensive chemotherapy treatments currently available. Because of their unique chemical reactions, metal peroxides hold enormous promise for medical applications. Their application in biomedicine involves cell imaging, drug delivery, and cancer therapy, which are associated with the productive toxic radicals produced from the reaction and the specific biochemical effects of the released metal ions [22,23,24]. Recently, many types of metal peroxide nanoparticles, including CuO_2_, CaO_2_, MgO_2_, ZnO_2_, BaO_2_, and TiOx, have gotten a lot of attention from researchers who want to find ways to treat cancer, stop bacterial infections, and help tissues grow back [25,26]. For a long time, researchers have conducted research to logically design drug-handling mechanisms, and have implemented various protocols to enhance targeting efficiency [27,28]. However, targeting strategies continue to fall short of achieving the desired outcome. The exploration of therapeutic interventions that are “disease-specific” via bio-orthogonal chemistry has attracted attention in the drug development field. To make diseases better, many nanoparticles have been suggested that can make or carry out chemical changes during treatment [29,30]. For instance, these “smart” drugs can detect the chemical environment of a disease and adjust their compound release accordingly, potentially delivering drugs to the body’s desired location. Metal ions and hydrogen peroxide groups present in the metal peroxides react with water to generate H_2_O_2_ and metal ions as byproducts [26]. A variety of biological activities could subsequently utilize the generated H_2_O_2_. For example, it serves as both the substrate and cofactor in a Fenton-like enzymatic process, resulting in large numbers of highly reactive •OH radicals [31]. Moreover, decomposing H_2_O_2_ can enhance the effectiveness of oxygen-consumption treatments such as photodynamic therapy (PDT) and radiation therapy [32,33]. Metal peroxides have proven useful in a variety of biological procedures, including both organic reactions and tissue regeneration [34,35]. Therefore, metal peroxide nanoparticles combine unique physical and chemical properties, reactive properties, and bioactivities to make new nanosystems that can be used in many biological ways. Figure 1 represents various metal peroxides that have sparked interest in this field, including copper peroxide (CuO_2_), calcium peroxide (CaO_2_), magnesium peroxide (MgO_2_), zinc peroxide (ZnO_2_), barium peroxide (BaO_2_), and titanium peroxide (TiOx). Researchers in many areas of biomedicine, such as catalytic nanomedicine, are looking into these metal peroxides right now. They are mainly interested in how they react with H_2_O_2_ and oxygen generators, as well as their bioactivity based on metal ions [36,37,38].

## 2. The Tumor Microenvironment

The TME is a complex and intricate system consisting of a variety of cellular and non-cellular elements [39]. Essentially, the TME encompasses everything within a solid tumor aside from the cancer cells themselves. Interactions between cells within the TME play a crucial role in influencing tumor development. These diverse cellular interactions ultimately dictate how tumors respond to various drugs [40,41]. There are several functionally distinct cells within the TME. Endothelial cells in the TME are critical in facilitating tumor growth and development while also protecting tumor cells from immune system assaults. Immune cells employ a variety of responses to combat cancer, with both innate and adaptive responses serving as the primary line of defense due to their diverse array of immune responses and activities that alter the tumor’s behavior in response to therapeutic interventions [42]. Therapeutic agents can activate innate immune cells such as natural killer cells (NKs), NKT cells, and T cells, which can directly lyse tumor cells or secrete cytokines and chemokines to inhibit tumor cell proliferation. In an adaptive response, T cells participate in immune responses to infiltrate the cancer cells’ response to drugs by blocking PD-1/PD-L1 [43]. All the healthy, non-cancerous cells within the tumor comprise the TME. These include fibroblasts, endothelial cells, neurons, adipocytes, adaptive immune cells, and innate immune cells. It also contains non-cellular components such as the extracellular matrix (ECM) and soluble substances such as chemokines, cytokines, growth factors, and extracellular vesicles. Crucially, the activation states and localizations of immune cells within the TME can lead to conflicting impacts on tumor growth. Fibroblasts within the TME play a significant role in the dissemination of cancer cells from the primary tumor site to the bloodstream, thereby enabling systemic metastasis. Furthermore, the extracellular matrix modulates the migration of cancer cells by altering their physical properties, composition, and topography. In addition to an extensive array of malignant cells, the TME also consists of secreted proteins, blood vessels, and non-malignant cells, all of which contribute to the sustenance and modulation of tumor development. It is possible that cytokines and growth factors made by either tumor cells or stromal cells in the TME may help with tumor growth that is not normal, the formation of new blood vessels, metastasis (the spread of cancer to other parts of the body), and resistance to treatment [44,45,46]. Various cellular components play a huge role in regulating tumor metastasis. Immune cells play a crucial role in altering the characteristics of cancer cells throughout various phases of the invasion–metastasis process, either by entering the tumor or by influencing the overall body environment [47]. As tumors progress, malignant cells devise tactics not just to dodge immune detection but also to trigger widespread reactions by taking advantage of certain immune cells, such as myeloid cells, to boost their ability to spread throughout the body [48]. Research suggests that the blood vessels within tumors exhibit changes in both structure and function, leading to hypoxia and a limited supply of nutrients. Moreover, hypoxia in the TME can alter gene expression in tumor cells, boosting their survival and resistance to apoptosis [49]. Tumors and tumor-associated macrophages (TAMs) release immunosuppressants in the TME. These include the checkpoint modulator PD-L1, which makes it harder for tumor-specific CD8+ T cells to kill cells [50]. Toll-like receptors (TLRs), which are found on macrophages and other cells in the TME, can also protect tumors by weakening the immune system. We now see TME cells and the molecules they release as essential in the development of cancer, making them promising targets for treatment. Tumor stage, cancer type, and patient characteristics can affect TME composition and activity. Consequently, different cells within the TME can either help or hinder tumor growth [51]. It is crucial to recognize that the specific makeup of the TME and its influence on cancer progression can differ greatly among patients and cancer types. The TME, together with the host immune system, is pivotal in shaping cancer progression and development. As a result, focusing on and altering the cells and factors within the TME during cancer treatment can aid in controlling cancer and enhance health outcomes [52]. Consequently, the pursuit of more efficient cancer treatment strategies, particularly for early-stage malignant tumors, is of paramount importance. In this context, Tang’s research team introduced an innovative method known as CDT in 2016 [53,54]. The disruption of intracellular redox equilibrium results in the production of free radicals. These free radicals are capable of causing oxidative damage and the death of DNA and other biomolecules. Consequently, this mechanism may lead to the eradication of cancer cells. Therefore, experts regard the Fenton reaction and Fenton-like reactions as highly effective methods for fighting malignant tumors [55]. In the list above, Fenton-like reactions include oxidation-reduction reactions that work like the normal Fenton reaction but do not use iron ions as a catalyst. These reactions frequently make use of a variety of metal ions or non-metal catalysts to enable comparable oxidation-reduction processes. Fenton-like reactions use transition metal ions such as cobalt, cadmium, copper, silver, manganese, nickel, and others to accelerate the production of ROS, crucial for tumor removal during cancer treatment [56,57,58,59,60].

### 2.1. Factors Associated with the TME

The TME plays a pivotal role in metastasis and cancer resistance, thereby presenting a substantial barrier to the clinical implementation of immunotherapy. Moreover, its critical involvement in proliferation, migration, and metastasis makes it an essential target in cancer treatment [61,62]. Considering the problems and restrictions of present treatment methods, changing the TME presents a promising alternative method that could greatly improve the effectiveness of present therapies. Furthermore, the TME is characterized by distinct features such as hypoxia, low pH, and an immunosuppressive milieu, which can serve as critical targets for its modulation [63,64,65]. Oxygen is essential for energy metabolism. Hypoxia leads to the creation of oxygen gradients within tumors, increasing hypoxia-inducible factor 1α (HIF-1α). This protein is a key indicator of hypoxia and plays a crucial role in the signaling pathways activated by it. Additionally, low oxygen levels can increase the production of HIF-1α. This can then raise the expression of programmed death ligand-1 (PD-L1) in cancer cells, making the immune system less effective [66]. On the other hand, a hypoxic TME makes it easier for tumors to grow and become resistant to drugs by causing abnormal angiogenesis, desmoplasia, and inflammation [67,68,69]. The primary regulators of gene transcriptional responses to hypoxia are HIF-1α and HIF-2α. These molecules stimulate the process of gene transcription, which in turn triggers hypoxic reactions. Additionally, they play a role in controlling the development of cancer and the reactions of the surrounding tissue. HIF-1α plays crucial roles in the functions of both the innate and adaptive immune systems, such as macrophages, neutrophils, dendritic cells (DCs), and lymphocytes. HIF-2α is linked to maintaining the balance of nitric oxide (NO) in macrophages. Taking away the aryl hydrocarbon nuclear translocator (ARNT)/HIF-1β gene from CD8+ T cells also lowers the production of molecules such as perforins and granzymes that kill cells [70]. HIF-1α has been documented to exhibit overexpression in multiple types of cancer and has been implicated in tumor survival by contributing to drug resistance [71]. Research has shown that HIF-1α is significantly present in many cancerous cells and plays a key role in inflammation, the development of tumors, and the ability of cancer cells to resist drugs when oxygen levels are low. In addition, they demonstrated the influence of hypoxia-inducible factor (HIF) in cancer stem cells. Furthermore, the hypoxic TME plays a crucial role in regulating and modulating p53 and mitochondria in cancer cells [72].

Nevertheless, hypoxia in the TME can create a low pH environment, resulting in multidrug resistance. Several factors play a role in this phenomenon, including genetic alterations, ion trapping, and the excessive activity of the multidrug transporter P-glycoprotein (P-gp). The semi-permeable nature of cellular membranes permits small, uncharged molecules to diffuse into cells, unlike charged particles [73]. Since many chemotherapeutic agents are pH-dependent, alterations in intracellular pH can hinder the diffusion of these drugs through the cell membrane, leading to drug resistance. Additionally, P-gp plays a crucial role in drug resistance within the low-pH TME [74].

The TME constitutes a diverse collection of unique cell types, encompassing immune cells, endothelial and inflammatory cells, fibroblasts, and lymphocytes. These cells are surrounded by the extracellular matrix (ECM), stroma, and blood vessels. They are also helped by the release of different proteins and the presence of organelles and chemokines [75]. The surrounding environment plays an important role in tumor advancement, spread, and recurrence. Immune cells are pivotal in differentiating between innate and adaptive immunity. Innate immunity is provided by cells such as macrophages and dendritic cells (DCs). The signaling pathways and chemokines in the TME determine whether these cells help or hurt the growth of tumors. Conversely, adaptive immunity is adept at accurately targeting tumor cells and is notably more successful in their elimination [75].

Each stage of cancer progression involves fibroblasts, which serve as the fundamental structural elements of the TME. Upon activation, we refer to these fibroblasts as cancer-associated fibroblasts (CAFs). Many important things happen in the TME because these cells make up important parts of the extracellular matrix (ECM) and release cytokines, growth factors, and matrix metalloproteinases that change the structure of the TME and help tumors grow and blood vessels form [75]. CAFs also block the activity of immune cells that can fight cancer and attract immune cells that help tumors grow [75]. By utilizing factors from the stroma, CAFs can transform environments that hinder cancer growth into ones that facilitate its spread. For instance, fibrinogen-like protein 2 can enhance the function of cancer-associated fibroblasts (CAFs) that promote cancer growth. This then increases the activity of myeloid-derived suppressor cells (MDSCs) through CXCL12, which helps the cancer spread [76]. Because of these different signaling pathways, the cancer-supportive TME has a disordered blood vessel system that makes oxygen levels drop [77]. These low-oxygen conditions can change the activity of the TME, stimulate cancer growth, and reduce the effectiveness of treatments [78]. By shielding factors that support cancer growth and the tumor itself, the TME, which influences drug work and absorption, can increase the cancer’s resistance to treatment. Because of this, the complicated web of interactions between cancer cells and the TME parts around them is always changing how fast tumors grow, how well treatments work, and how drug resistance develops. Identifying the profound effects of these interactions, we focus on the aberrant components of the TME as a viable approach for managing solid tumors, particularly those characterized by aggressiveness and treatment resistance, such as those observed in the gastrointestinal (GI) tract.

Regimens designed to regulate the immune system employ therapeutic strategies, including pembrolizumab, which targets programmed cell death protein-1. However, these treatments have shown limited effectiveness in treating gastrointestinal cancers, which include colorectal and pancreatic cancer (CRC and PC), due to the intricate nature of the TME [79,80]. Additional approaches encompass chimeric antigen receptor T (CAR-T) cell therapy, which has shown modest success in the treatment of lung cancer and melanoma [77]. The modest advancements in cancer treatment have prompted researchers to investigate alternative strategies to augment the effectiveness of both solid and targeted therapies [81]. Nanoparticles (NPs) have emerged as a promising tool for improving immune therapies due to their extended presence in the body and their ability to target specific areas, thereby reducing harmful effects [82]. NPs are adaptable substances that can target key components of the TME to alter the immunosuppressive setting. For example, the growth of tumors and their swift expansion result in reduced oxygen levels and immunosuppression. Regulatory T cells and myeloid-derived suppressor cells (MDSCs) facilitate this process. Subsequently, these cells are responsible for the production of vascular endothelial growth factor (VEGF) and transforming growth factor beta (TGF-β) [83,84,85]. These substances inhibit the function of dendritic cells and promote the growth of tumor-supportive macrophages, resulting in abnormal scarring. Tailored nanoparticles (NPs) can precisely target certain parts of the TME. This changes the immunosuppressive environment into one that supports the immune response and makes cells more receptive to immunotherapy. These nanoparticles act as passive carriers for drugs, enabling them to remain in the body for extended periods. The enhanced permeability and retention (EPR) effects, irregular lymphatic drainage, and permeability of blood vessels can cause them to accumulate within malignant tissue compared to adjacent normal tissue. Furthermore, by modifying their structure and binding them to specific molecules, we can selectively direct them to specific regions, thereby enhancing their efficacy, compatibility, and efficiency [86,87]. Consequently, we can precisely engineer and utilize nanomaterials, characterized by their diverse chemical compositions, shapes, and sizes, as effective combinations to administer targeted therapies to distinct areas within the TME. The Food and Drug Administration (FDA) has granted regulatory approval for the clinical use of nanodrugs, including Abraxane and Doxil [88].

Taking these factors into account, focusing on the acidic and hypoxic conditions of the TME can be highly effective. For example, one approach could be to inhibit VEGF or the PI3K/AKT/HIF-1α pathway to target hypoxia [72]. Overall, it is very important for therapeutic interventions to accurately target key TME features, such as hypoxia-induced acidity, tumor-associated macrophage (TAM) reprogramming, extracellular matrix (ECM) remodeling, and vasculature remodeling, in response to hypoxia. However, there are significant challenges in precisely delivering and targeting these treatments. Nanotechnology can overcome these challenges. Figure 2 represents the characteristic features of the TME, such as low pH, hypoxia, and an immunosuppressive environment.

### 2.2. Targeting the Tumor’s Microenvironment

The previous section detailed the potential of TME modulation as a key treatment option for different solid cancer types. This section emphasizes the impact of NP-based nanotechnology on the TME. Nanoparticles (NPs) are typically defined as particles measuring approximately 1–100 nm, exhibiting a range of characteristics and properties that differ significantly from those of their bulk counterparts. This distinction leads to unique properties, primarily due to their elevated surface-to-volume ratio [89]. For an extended period, researchers have recognized the antibacterial efficacy of silver nanoparticles [90,91]. Nevertheless, in recent decades, advancements in technology have enabled the production of a diverse array of nanoparticles, including quantum dot nanoparticles and nanofibers.

Various studies have explored the use of nanoparticles for targeted drug delivery across different diseases, leading to promising clinical outcomes [92,93]. Nanoparticles’ unique shapes and high surface area-to-volume ratio make it easier to package medicines in a variety of places, such as on the outside, in capsule-like structures, and on the inside, in dendrimers. This process leads to the efficient modulation of the TME [94]. The enhanced permeability and retention (EPR) effect plays a vital role in drug accumulation within cancer. This effect works especially well for nanoparticles (20–200 nm) because they can change sizes so easily. This lets them target cancer cells by getting through the pores in vascular endothelial cells and making them more permeable [95].

While research has shown that EPR could be very helpful for concentrating nanoparticles inside cancer cells, it has been hard to get the nanoparticles to where they need to go because cancer cells are very different and the blood vessels inside cancerous tissues are very complicated. Therefore, scientists have been working on active-targeted drug delivery strategies, even though they know that the enhanced permeability and retention (EPR) effect seen in these systems is mostly passive. This methodology engineers nanoparticles for selective delivery.

This methodology engineers nanoparticles to selectively interact with markers exclusive to cancer cells. The heightened antigenicity of cancer cells highlights the significance of the VEGF receptor as a pivotal target for active targeting. Concentrating on this receptor and simultaneously exploiting the EPR effect can significantly enhance the accumulation of therapeutic agents within the TME. Moreover, the ability of other therapeutic agents targeting cancer cells, such as anginex and RGD peptides, to bind to galectin-1 and integrin v3, respectively, sets them apart [96,97,98].

Additionally, the production of ROS through nanoparticles results in the targeting of the TME. The Fenton-like reaction transforms H_2_O_2_ into more harmful OHs. •OHs, as active single electron molecules, exhibit a potent oxidizing ability, capable of oxidizing the amino groups in proteins and polyunsaturated fatty acid chains in lipids, while also inflicting damage on DNA. Several studies suggested that ROS interact with the HIF1 pathway and alter HIF1α expression in cancer cells. H_2_O_2_ changes ferrous iron (Fe^2+^) into ferric iron (Fe^3+^), which stops ferrous iron from attaching to the prolyl hydroxylases (PHDs), which are enzymes that change HIF1α into a different form. Additionally, studies have demonstrated that ROS influence the HIF1 pathway by altering the levels of H_2_O_2_, Fe^2+^, Asc, 2OG, or SC [99].

Metal nanoparticles (NPs) serve not only as drug delivery systems but also as therapeutic agents. The Plasmon resonance and photoluminescence characteristics of these metal NPs are noteworthy in this context [100]. Additionally, a variety of studies have used NPs to modify the TME in a number of ways, including by modifying the TME’s acidity, changing the tumor ECM’s structure, addressing the TME’s immunosuppressive elements, and reducing tumor hypoxia through techniques such as increasing oxygen delivery, improving oxygenation, and lowering oxygen consumption [101].

### 2.3. The Strategy Involves the Surface Modification of Nanoparticles

In the realm of drug delivery via nanoparticles, there are two main approaches to targeting: passive and active targeting. Passive targeting uses the EPR effect to get nanoparticles into the bloodstream through tumors with leaky blood vessels and buildup in the tissue around the tumor [102]. This phenomenon occurs due to the peculiar morphology of tumor blood vessels, rendering them more permeable in comparison to their counterparts in healthy tissues. Consequently, this increased permeability facilitates the accumulation of nanoparticles. Nonetheless, the extent of the EPR effect differs among various tumors, and it does not provide sufficient specificity on its own [102,103].

Conversely, active targeting entails the coating of nanoparticles with specific molecules capable of identifying and binding to molecules overproduced on the surface of target cells [104]. This can encompass molecules such as antibodies, aptamers, peptides, carbohydrates, and small molecules selected for their selectivity [105]. Unlike passive targeting, which depends on general tumor characteristics, active targeting employs molecular recognition to deliver drugs with greater precision [106]. However, creating and applying active targeting strategies is not without its difficulties, such as finding the right combination of target and binding molecules, ensuring the right amount of binding molecules, and avoiding potential immune reactions [105].

Despite these challenges, active targeting has demonstrated greater efficacy in treating diseases during early animal studies compared to passive targeting, which relies on the EPR effect [107]. Targeting nanoparticles to cancer cells has recently advanced through the use of overproduced specific molecules such as transferrin, folate, epidermal growth factor, and prostate-specific membrane antigen [108,109,110,111]. Additionally, we can target active molecules at overproduced receptors on tumor blood vessels.

Researchers are exploring a new strategy that combines passive targeting for initial body entry and active targeting for precise delivery to specific cells. Experts regard the dual passive and active targeting approach as a promising strategy to enhance the delivery and efficacy of nanoparticles. We anticipate further advancements in nanoparticle applications for drug delivery as the fields of material science, bioconjugation, and molecular targeting continue to progress [112].

Functional groups such as thiols and amines were employed to covalently link drug molecules or targeting ligands to the exterior of nanoparticles. This process, known as bioconjugation, significantly enhances the targeting abilities of nanoparticles. Sheng et al. have developed nanoparticles coated with calcium peroxide and a pH-responsive methacrylate-based co-polymer. The design of this co-polymer incorporates a tertiary amine to produce oxygen in the hypoxic tumor environment [113].

## 3. Fenton Reaction

Recently, Fenton reaction-based catalytic nanoparticles have become a new way to treat tumors that are very selective [114,115]. Generally, oxidative therapy applies Fenton chemicals to induce an asymmetric reaction and transform hyperexpressed H_2_O_2_ in tumors into harmful hydroxyl radicals [116,117,118]. This section systematically presents the treatment mechanisms of Fenton reactions. The field of water treatment and pollution control has significantly advanced thanks to the invention of Fenton/Fenton-like reactions by Henry J. Fenton in the late 19th century. Moreover, researchers have identified and applied chemodynamic therapy (CDT) in cancer chemotherapy, including Fe, Cu, Mn, Mo-, Cr-, Ti-, Co-, Al-, Ce-, and Ru-based nanomaterials [119,120,121,122,123,124,125,126] (Figure 1). Fundamentally, CDT usually depends on interactions between Fe^2+^ and H_2_O_2_.
Fe^2+^ + H_2_O_2_ → Fe^3+^ + •OH + OH^−^,(1)

Both the Haber–Weiss reaction (˙O_2_^−^ + H_2_O_2_ → OH + OH^−^ + O_2_) and the Haber–Weiss chain reaction (O_2_^−^ + H_2_O_2_ + H^+^ → OH + H_2_O + O_2_) produce highly oxidative OH, which damages biomolecules in many ways, including lipid peroxidation and DNA damage [127]. These reactions, in general, are rather complex and sustain a sequence of reactions, including the beginning, spreading, and ending of reaction components. Specifically, according to the Fenton reaction mechanism, OH is initially generated through the reaction between Fe^2+^ and H_2_O_2_, as depicted in Equation (1). The antioxidant GSH then reduces the generated Fe^3+^ in the TME to replicate Fe^2+^, resulting in a replenishment of Fe^2+^.
Fe^3+^ + GSH → Fe^2+^ + GSSG (glutathione disulfide),(2)

Owing to its distinct pattern of producing extremely toxic •OH, CDT can overcome the major challenges posed by hypoxia resistance, penetration depth limitation, and the harm that chemotherapy and radiation therapy inflict on healthy cells [128]. Furthermore, it should be mentioned that the Fenton reaction produces •OH (E (•OH/H_2_O) = 2.8 V), which is greater and more resilient than the ^1^O_2_ (^1^O_2_/H_2_O) = 2.17 V) produced by PDT or SDT [129]. However, the limited H_2_O_2_ concentration and excessive GSH expression somewhat restrict the healing impact of traditional Fenton reaction-based CDT. As a result, a novel approach based on Fenton reaction-based CDT has been proposed as a promising alternative strategy for combating cancer [57]. Furthermore, notable developments have also been accomplished in multimodal cooperative treatment mediated by CDT [130,131,132,133,134,135]. Due to the Fenton reaction’s low efficiency in treating malignant tumors and its limited ability to generate •OH, effectively eliminating cancer cells remains a challenge. To tackle this issue, researchers have developed a variety of methods to catalyze the Fenton reaction and enhance its therapeutic effectiveness against malignant tumors. Possible limitations include the abundant intratumoral H_2_O_2_ for •OH production, the mild acidity of the TME that is not optimal for the Fenton reaction, and the tumor’s own anti-oxidative systems that scavenge ROS. To address these multimodal systems, it is suggested that CDT be combined with other complementary therapeutics in synergistic tumor therapy to combat the TME. The performance of CDT is constrained by the poor performance of the TME, and neither the exogenous H_2_O_2_ nor the acidity within the cancer cell is sufficient to trigger the traditional Fenton reaction. Additionally, the intratumoral overexpressed GSH neutralizes the Fenton-induced ROS through antioxidant mechanisms [135,136].

This review discusses the application of metal peroxide-based nanocatalytic medicines in chemotherapy CDT and the associated synergistic therapy strategies (Figure 2). It provides a thorough understanding of this emerging paradigm in cancer treatment and also addresses the potential hurdles and challenges that may arise in the future advancement of CDT for clinical translation.

### Role of Chemodynamic Therapy in Cancer

CDT is a new, minimally invasive method that changes harmful H_2_O_2_ into the much more dangerous hydroxyl radical (•OH) through Fenton or Fenton-like reactions [116]. The OH can induce significant cell death in cancerous cells by damaging DNA, deactivating proteins, and initiating phospholipid membrane peroxidation, as represented in Figure 3 [136]. CDT offers several advantages due to its selectivity; as it is more effective in environments with higher levels of H_2_O_2_ (such as in tumors compared to normal tissues), damage to healthy tissues is reduced. Consequently, CDT provides several benefits over traditional treatment options, including minimal invasiveness, high selectivity, and fewer side effects. The natural concentration of H_2_O_2_ typically ranges from 10 to 50 μM [137]. However, this concentration alone is not enough to produce the necessary quantity of hydroxyl radicals for the successful operation of CDT. Therefore, it is crucial to develop innovative strategies to boost the levels of H_2_O_2_ within tumors. This would improve the generation of hydroxyl radicals through Fenton-like reactions, thereby enhancing the effectiveness of CDT [138]. Utilizing biochemical processes is a method to increase the endogenous levels of H_2_O_2_ in tumors [139]. Two of the main ways this is done are through enzyme catalysis, especially with glucose oxidase (GOx) and superoxide dismutase (SOD). GOx speeds up the reaction involving water, oxygen, and glucose, resulting in the formation of gluconic acid and H_2_O_2_. On the other hand, SOD facilitates the generation of H_2_O_2_ from superoxide anion radicals [140]. These processes effectively elevate the H_2_O_2_ content in tumors by catalytic reactions, utilizing chemicals present within the tumor. Metal peroxides have the potential to generate oxygen or serve as a reaction substrate to combat tumor hypoxia, thereby enhancing the availability of oxygen for chemotherapy drugs and improving the efficacy of chemotherapy [141]. Due to their negligible adverse effects on normal tissues within a living organism, metal peroxides, which decompose under the acidic conditions of the TME to yield metal ions and H_2_O_2_, represent a promising alternative source of H_2_O_2_ [142]. Upon exposure to the acidic environment present within TMEs, transferrin-modified magnesium oxide nanosheets exhibit a swift and significant generation of H_2_O_2_. Subsequently, these nanosheets engage in a Fenton reaction, facilitated by the release of metal ions from transferrin. This reaction results in a notable enhancement in the production of toxic hydroxyl radicals (•OH), thereby contributing to the efficacy of cancer therapy [143].

The application of MnO_2_ nanoparticles as a novel chemodynamic approach to augment cancer treatment via CDT has been extensively investigated. Upon cellular absorption, MnO_2_ interacts with intracellular glutathione (GSH), culminating in the formation of oxidized glutathione (GSSG) and manganese ions (Mn^2+^). These manganese ions demonstrate pronounced Fenton-like catalytic activity, which aids in the production of highly reactive hydroxyl radicals from endogenous H_2_O_2_ in the presence of physiological bicarbonate (HCO_3_−) ions. As depicted in the Figure 3, the reduction in GSH levels compromises the antioxidant defense system (ADS), rendering cancer cells more vulnerable to the hydroxyl radicals produced via the Mn^2+^-mediated Fenton-like reaction. Consequently, this enhances the cell death pathway CDT, leading to the eventual demise of the cells [144].

## 4. Metal-Based Peroxides for Cancer Therapy

### 4.1. Copper Peroxide

Copper is pivotal in the metabolic processes of both flora and fauna. It is distinguished by its soft, malleable characteristics, alongside its exceptional thermal and electrical conductivity. In comparison to metals such as platinum, silver, and gold, copper nanoparticles present a more economically viable alternative among the transition metals under investigation. The capability to formulate Fenton nanoagents for the development of catalytic nanotherapeutics is enhanced by the ability of copper peroxides to produce H_2_O_2_. Multifunctional copper peroxide nanodots have been successfully synthesized in an aqueous environment through the amalgamation of copper chloride, H_2_O_2_, and sodium hydroxide [145].

Polyvinylpyrrolidone (PVP) was utilized in this process due to its ability to provide the necessary surface functionalization to ensure the exceptional stability of nanodots within a physiological environment while also controlling the diameter of the nanodot particles. Their approximate 5 nm particle size facilitates efficient tumor accumulation [146,147]. Upon the reaction of the CuO_2_ nanodots with water, a chemical transformation was initiated, culminating in the production of H_2_O_2_. Furthermore, the presence of Cu^2+^ catalysts facilitated a Fenton-like reaction, leading to the generation of highly reactive hydroxyl radicals. It is noteworthy that H_2_O_2_ was the sole reactant in this process [148]. The produced hydroxyl radicals led to lysosomal lipid peroxidation, which in turn induced lysosomal membrane permeabilization, which ultimately resulted in cancer cell death [149,150]. A highly efficient nanocomposite, referred to as UCN@CuO_2_-GOx (UCCuG), was successfully synthesized. This synthesis facilitated the enhancement of starvation therapy and the self-sustaining decomposition of tumor cells in vitro through the incorporation of glucose oxidase (GOx) within the nanocomposite. The GOx within the nanocomposite utilized glucose for starvation therapy upon the entry of UCCuG into tumor cells. Furthermore, the acidic environment of the lysosome stimulated the liberation of copper ions (Cu^2+^) and H_2_O_2_ during the decomposition of UCCuG. Subsequently, copper ions were reduced to copper ions (Cu+) within tumor cells, and subsequently, copper ions catalyzed the release of H_2_O_2_, thereby generating hydroxide ions (•OH) for CDT. The results obtained from in vitro experiments were exceptionally promising, indicating the significant enhancement of CDT through starvation therapy [151]. The development and design methodology of sorafenib-loaded copper peroxide nanoparticles (CuO_2_-PVP-SRF NPs) was conducted with meticulous precision, with the primary objective of augmenting the cytotoxic efficacy of CDT against cancer cells. The SRF-mediated system dysfunction, coupled with the role of Cu^2+^ in redox reactions, can lead to the depletion of intracellular glutathione (GSH), thereby inactivating glutathione peroxidase (GPX4) and facilitating the production of copper ions (Cu+). In scenarios where the reducing capacity is compromised, a considerable amount of hydroxyl radicals (•OH) is generated through a Cu+-mediated Fenton-like reaction. This, in turn, results in cell apoptosis and the accumulation of lipid hydroperoxides, which subsequently induces ferroptosis [152]. Ma et al. have delineated the synthesis process of self-assembled copper-cysteine mercaptide nanoparticles (Cu-Cys NPs), which function as copper-rich Fenton nanocatalysts for applications in antitumor therapy [153]. These Cu-Cys nanoparticles exhibit stability under physiological conditions, yet they demonstrate a swift release of Cu^2+^ upon internalization by cancer cells. Within the cellular milieu, Cu^2+^ is initially transformed back into Cu+ through oxidation-reduction reactions, which are catalyzed by intracellular glutathione (GSH). Subsequently, Cu+ can catalyze a Fenton-like reaction with the intracellular H_2_O_2_, leading to the formation of hydroxyl radicals (•OH). In vivo, studies have indicated that Cu-Cys NPs possess a tumor suppression efficacy of approximately 72.3%, which is notably higher than the 17.1% inhibition rate observed with the chemotherapy agent doxorubicin (DOX). Despite the increased expression of intracellular H_2_O_2_ in various tumor cell types compared to normal cells, the endogenous levels of H_2_O_2_ are insufficient to achieve optimal outcomes in chemotherapy-based CDT, as shown in Figure 4A. Liu and colleagues have published a study on a versatile nanoplatform that combines hypoxia relief and ROS enhancement for synergistic cancer treatment using CDT/PDT [154]. The study involved the synthesis of cancer cell membrane-coated mesoporous copper/manganese silicate nanospheres (mCMSNs). These synthesized mCMSNs were found to generate oxygen, which, when subjected to 635 nm laser light, could be converted into toxic oxygen species (^1^O_2_), thereby augmenting the effectiveness of photodynamic therapy (PDT). Moreover, the presence of glutathione (GSH) facilitated the biodegradation of mCMSNs, resulting in the concurrent release of Fenton-like copper (Cu^+^) and manganese (Mn^2+^) ions, which subsequently generated hydroxyl radicals (•OH). The research demonstrated a significant reduction in tumor hypoxia following the administration of mCMSNs. The results also confirmed the production of •OH and ^1^O_2_ through the collaborative action of chemotherapy/PDT. The study suggested that the combination of mCMSNs and exposure to 635 nm laser light exhibited enhanced synergistic effects in tumor suppression. However, it was observed that neither PDT (635 nm laser) nor CDT (mCMSNs) alone was effective in halting tumor growth. Furthermore, the study highlighted the use of endogenous substances to catalyze oxygen production and the significant consumption of GSH for the precise regulation of the hypoxic TME. Liu et al. developed biocompatible copper ferrite nanospheres (CFNs) characterized by a relatively narrow bandgap, serving as a comprehensive nanoplatform for synergistic therapy involving CDT, PDT, and PTT [155]. In this groundbreaking system, complex fluorescent nanoparticles (CFNs) are engineered to incorporate two distinct redox pairs: Fe^2+^/Fe^3+^ and Cu^+^/Cu^2+^. This configuration facilitates the production of a considerable amount of •OH through Fenton or Fenton-like reactions. Moreover, CFNs play a pivotal role in elevating oxygen levels by catalyzing the decomposition of H_2_O_2_ and depleting glutathione (GSH). Functioning as proficient photosensitizers, CFNs are instrumental in the transformation of oxygen to •O_2_, which is crucial for Photodynamic Therapy (PDT) applications. As a result, cytotoxic •OH and •O_2_ are generated via a photogenerated electron/hole (eCB−/hVB+) pair within the CFNs, thereby enhancing the effectiveness of chemotherapy-drug therapy and PDT. Additionally, CFNs demonstrate remarkable photothermal properties upon exposure to 808 nm laser light. Imaging studies (DCF) reveal that HeLa cells treated with CFNs and subjected to 650 nm laser light exhibit a higher concentration of reactive ROS compared to untreated cells, underscoring the significant ROS production that occurs through the synergistic interaction of photo-enhanced CDT and PDT. Furthermore, the generated •O_2_ suppresses green fluorescence (RDPP), and GSH is nearly completely consumed following CFN treatment, illustrating CFNs’ capacity to effectively alter the TME. The tumor temperature rises to 57 °C within 10 min following 808 nm laser irradiation, in contrast to other groups that do not show significant heating, demonstrating CFNs’ efficacy as efficient photothermal agents for localized heat generation. The combined impact of CFNs, in conjunction with treatments with 650 nm and 808 nm lasers, results in the complete eradication of tumors, a feat that is unattainable through bimodal Photodynamic Therapy/Photodynamic Therapy/Chemotherapy (PTT/PDT/CDT) synergistic therapies. A newly engineered silica nanoplatform, designated CuO_2_/DDP@SiO_2_, merges copper and cisplatin to interfere with copper metabolism, thereby augmenting chemotherapy and chemotherapy-based CDT. This combination yields a synergistic approach to cuproptosis and chemotherapy/chemodynamic anticancer therapy [156].

The creation of oxygen, the rise in acidity, and the breakdown of cellular glutathione (GSH) cause a phenomenon called cuproptosis. This leads to the clumping of proteins attached to lipoproteins inside the cell, which are the main victims of copper-induced damage. In vitro studies have confirmed that a reduction in GSH’s binding to cisplatin significantly increases the concentration of cisplatin within cells. Furthermore, copper oxide (CuO_2_) has been shown to markedly diminish the expression of multidrug resistance-associated protein 2 (MRP2) by inhibiting the O_2_-dependent activation of hypoxia-inducible factor 1 (HIF-1). As a result, the entire pathway for cisplatin efflux is disrupted, thereby augmenting the anticancer efficacy of cisplatin in both in vitro and in vivo models. In addition, Huang et al. developed nanocomplexes containing CuO_2_ loaded into the generation-5 poly(amidoamine) dendrimer for the delivery of p-carboxybenzenesulfonamide (BS) and iron (Fe)–tannic acid (TF) complexes for targeted MR imaging and enhanced ferroptosis/cuproptosis/CDT against triple-negative breast cancer cells, as shown in Figure 4B [157]. The TME of cancer cells was regulated due to these nanocomplexes due to the overexpressed CA IX in cancer cells. Therapeutic efficiency was achieved by inhibition of CA IX activity, which accelerates Fe^3+^/Cu^2+^ release, and also due to the Fenton reaction.

### 4.2. Calcium Peroxide

Calcium peroxide (CaO_2_) is widely employed in the treatment of tumors owing to its capacity to generate H_2_O_2_ and oxygen (O_2_) within the TME without the necessity of external stimuli. Furthermore, it can rapidly discharge a substantial amount of free calcium ions, which expedites cell demise by augmenting oxidative stress through calcium excess and mechanisms induced by H_2_O_2_ [158,159,160,161,162]. It is gratifying to observe that the cytotoxic impact facilitated by calcium oxide (CaO_2_) is not confined to specific tumor categories. This mechanism of action can be applied to a broad spectrum of tumors, encompassing those of the liver, colon, lung, and breast. Conversely, normal cells demonstrate a heightened susceptibility to adverse effects, thereby protecting them from harm in comparison to tumor cells [141,163]. Moreover, calcium peroxide (CaO_2_) experiences accelerated hydrolysis within the acidic TME, leading to a swift accumulation of intracellular calcium and an escalation of oxidative stress. The susceptibility of Calcium Oxide (CaO_2_) to acidic conditions primarily stems from the role of acidic environments in facilitating the generation of protons during the process of water ionization. The decomposition rate of calcium peroxide is influenced by the concentration of nearby protons, culminating in the formation of calcium ions (Ca^2+^) and H_2_O_2_ as the final products. The presence of H_2_O_2_ induces severe oxidative stress within cells, thereby disrupting the normal functioning of intracellular calcium channels. This disruption hampers the removal of excess calcium ions from cells and complicates the maintenance of intracellular calcium levels within the normal range. Consequently, this leads to an accumulation of calcium ions, which can result in cell necrosis [164]. Since the chemically inert fraction of CaO_2_ nanoparticles cannot trigger chemical reactions, it is necessary to combine them with other Fenton chemicals to achieve therapeutic goals [165]. which resulted in Fenton-based tumor killing and H_2_O_2_ self-supply. Furthermore, within tumor cells exhibiting a notable reduction in catalase activity, the ensuing temporary elevation in oxidative stress will precipitate the degradation of proteins, thereby impairing the function of calcium ion channels. This impairment facilitates the uncontrolled buildup of intracellular calcium, which interferes with the normal propagation of calcium signals, ultimately culminating in cell demise. However, the pronounced instability of CaO_2_ leads to the liberation of H_2_O_2_ and calcium ions (Ca^2+^) into the bloodstream, resulting in adverse effects on the body. This constraint impedes its practical application. As a result, the emphasis of current therapeutic strategies utilizing CaO_2_ is directed toward reducing its non-specific toxicity and improving its stability within the bloodstream [159,166,167]. It is noteworthy that substantial research findings have been presented. For instance, Liu and his colleagues have delineated a hyaluronic acid (HA)-modified calcium oxide (CaO) and copper oxide (CuO) nanocomposite that successfully postpones peroxide hydrolysis under normal physiological conditions. Furthermore, this composite has been demonstrated to exhibit a synergistic antitumor effect by combining multiple metal peroxides, which is represented in Figure 5A [159]. Sodium hyaluronate-modified nano-CaO_2_ nanoparticles demonstrate stability within the bloodstream until they encounter the TME, at which point the protective coating is compromised by the overproduction of hyaluronidase by the TME, thereby releasing the drug. Upon accumulation in the TME, a considerable release of H_2_O_2_ occurs, initiating a Fenton reaction. This reaction involves the interaction between the released copper ions and the generated H_2_O_2_, leading to the formation of a substantial number of hydroxyl radicals. These radicals, in turn, augment the amplification of transient oxidative stress within cells. Furthermore, the nanoparticles can respond to signals from the TME, facilitating the cleavage of the protective coating. Additionally, external stimuli can serve as a trigger for this response. For example, Liu et al. have developed a calcium-based nanoparticle, referred to as (MSNs@CaO_2_-ICG)@LA, which is characterized by the self-supplied mixture of H_2_O_2_ and oxygen (O_2_). This nanoparticle is constructed from a combination of manganese silicate (MSN)-loaded nano-CaO_2_ and indocyanine green (ICG). Additionally, the nanoparticle has been further modified by incorporating lauric acid (LA) onto its surface. The incorporation of lauric acid is noted to occur at a phase change point ranging from approximately 44 to 46 degrees Celsius, which is represented in Figure 5B [167]. In the presence of an 808 nm laser, indocyanine green (ICG) exhibits the capability to generate singlet oxygen (^1^O_2_) concurrently with the production of elevated temperatures, which are sufficient to melt lauric acid (LA) and subsequently release calcium oxide (CaO_2_). The authors have focused on the synergistic generation of ROS within the calcium peroxide system to minimize resource utilization. This strategy integrates the production of H_2_O_2_ via calcium oxide (CaO_2_), H_2_O_2_-mediated chemiluminescence CDT, and oxygen-mediated photodynamic therapy (PDT), all of which are part of an open-source approach. Additionally, the use of glutathione (GSH) facilitated by MSN serves as a strategy to mitigate the consumption of ROS, thereby protecting them from elimination. This drug delivery approach has demonstrated significant tumor-suppressive effects in both in vivo and in vitro environments, thereby enhancing the cancer treatment strategy by leveraging ROS from multiple perspectives. Calcium Oxide (CaO_2_) stands out as a significant advancement in the development of calcium-based materials for tumor therapy, as it combines the dual functions of calcium (Ca^2+^) and H_2_O_2_ to induce cell apoptosis while also providing valuable insights for the advancement of “green tumor therapy.” His team has developed a liposome-coated CaO_2_ nanocarrier for the simultaneous delivery of a chemotherapy drug, doxorubicin (DOX), and a biocompatible Fenton catalyst, an iron–oleate complex [168]. The nanoparticles, encapsulated within solid lipid shells, experienced disruption upon encountering cancer cells that exhibited an elevated expression of lipase. This interaction led to the liberation of iron-oleate into the cellular cytoplasm. Moreover, the nanoparticles’ calcium cores were sensitive to the acidic nature of the aqueous environment, resulting in the production of H_2_O_2_ and the subsequent release of doxorubicin into the cytoplasm. The Fe^3+^ ions derived from the iron oleate played a crucial role in facilitating the production of oxygen, which is beneficial for chemotherapy treatments that are less affected by hypoxia. Additionally, the formation of hydroxyl radicals enhanced the effectiveness of cancer treatment strategies based on CDT. Collectively, these processes synergistically increased the effectiveness of the nanoparticles in combating cancer cells. Mesoporous calcium peroxide nanoparticles, characterized by their hollow structure, were synthesized employing a straightforward hydrolysis-precipitation technique for the treatment of tumor calcicoptosis [169]. Hyaluronic acid plays a pivotal role as a stabilizer during the synthesis of HMCPN in the presence of ultrasound. Subsequently, it accumulates within tumors due to the enhanced permeability and retention (EPR) effect, leading to its degradation in the acidic conditions prevalent within the TME. This degradation process liberates significant amounts of calcium ions (Ca^2+^) and H_2_O_2_. The increased levels of intracellular calcium ions stimulate the release of extracellular calcium ions, which, in turn, contribute to the calcification of cell membranes and the onset of calcicoptosis. Moreover, the surplus of intracellular calcium ions can precipitate mitochondrial dysfunction, which, in consequence, diminishes the activity of matrix metalloproteinases (MMPs). The accumulation of calcium ions within the mitochondria disrupts the tricarboxylic acid cycle, thereby hindering the acceptance of electrons by cytochromes in the respiratory chain. This disruption leads to the self-reduction of oxygen (O_2_) and the formation of superoxide anions (O^2•−^). The elevated levels of intracellular H_2_O_2_ intensify oxidative stress, culminating in the process of lipid peroxidation (LPO). Zhang and his team have engineered a liposome-based nanoplatform designed for the implementation of dual-stage light-driven photodynamic therapy (PDT). Within this nanoplatform, hydrophilic polysulfated glycosaminoglycans (PS, including methylene blue, MB), and calcium oxide nanoparticles (CaO_2_ NPs) are encapsulated within both an aqueous cavity and a hydrophobic layer, respectively. Upon reaching the tumor tissue, the CaO_2_ nanoparticles within the liposomes undergo a reaction with water, resulting in the production of oxygen within the mildly acidic TME. This process is intended to mitigate tumor hypoxia. Subsequently, brief irradiation is administered in the first stage, which activates singlet oxygen (^1^O_2_) through the interaction with MB, subsequently leading to the rupture of the liposomes and exposing CaO_2_ to water once more, thereby generating additional oxygen. The final stage involves prolonged irradiation, which significantly augments the efficacy of the photodynamic therapy in an oxygen-rich TME [157]. This innovative dual-phase light strategy, utilizing calcium oxide dioxide (CaO_2_), is carefully engineered to enhance the capacity for oxygen delivery, facilitated by CaO_2_. It functions as a nanoscale structure, presenting considerable potential for addressing tumor hypoxia and impeding tumor metastasis. The approach encompasses the encapsulation of a hydrophobic aza-BODIPY dye (B1), oxygen-producing CaO_2_, and hydrophilic ammonium bicarbonate (NH_4_HCO_3_) within phospholipid-coated liposomes, which has been successfully demonstrated for the development of a self-sustaining oxygen photodynamic therapy (O_2_ PDT) treatment [78]. In the course of this study, NH_4_HCO_3_ is identified as a molecule that demonstrates a reaction to changes in temperature. When exposed to near-infrared (NIR) radiation, the liposome system experiences an increase in temperature, which is attributed to the B1 component. At a temperature of 40 °C, NH_4_CO_3_ undergoes thermal decomposition, resulting in the production of CO_2_. This process leads to the expansion of the liposomes, which, in turn, facilitates the complete reaction between CaO_2_ and CO_2_, thereby enhancing the photodynamic therapy (PDT) effect of B1. In summary, in vivo, experiments were conducted to assess the effectiveness of tumor treatment, and it was demonstrated that the optimal production of O_2_ from the CaO_2_/B1/NH_4_HCO_3_ liposomes was advantageous for the generation of O_2_ in the presence of the photosensitizer B1. This resulted in the inhibition of tumor growth and, in certain instances, the potential eradication of tumors [170]. In addition to the previously mentioned examples of oxygen generation being activated by light, it has been observed that pH levels also serve as a potential trigger. The study carried out by Callan et al. involved the encapsulation of CaO_2_ particles with a methacrylate-based co-polymer that is sensitive to changes in pH [113]. In an acidic environment, the tertiary amine segment of the co-polymer is rendered ionized, thereby rendering CaO_2_ and enhancing the production of oxygen. The TME fosters an environment conducive to the dissolution of these co-polymers.

### 4.3. Other Metal Peroxide Nanoparticles

Magnesium-based nanoparticles are extensively acknowledged for their applications within biological environments. Specifically, magnesium peroxide nanoparticles have been documented for their potential in cancer treatment. Furthermore, magnesium peroxide has been utilized in the development of metal-organic frameworks (MOFs). These frameworks have been engineered for photodynamic therapy (PDT) to generate sufficient oxygen to mitigate hypoxia and enhance the effectiveness of PDT [24]. A metal-organic framework (Hf-MOF-MgO_2_/DNA) was employed, in conjunction with magnesium peroxide nanoparticles, to augment the efficacy of photodynamic therapy (PDT). This composite platform demonstrated the capability to mitigate the challenges posed by hypoxia by generating an adequate supply of oxygen, thereby augmenting the effectiveness of PDT. Furthermore, the aptamer component of this platform exhibited specificity towards tumor cells, facilitating targeted therapy. In vitro PDT treatments utilizing the Hf-MOF-MgO_2_/DNA platform, specifically targeting 4T1 cells, yielded a survival rate of 15.15% for these cells, in comparison to a survival rate of 28.69% for A549 cells under identical conditions. Titanium peroxide nanoparticles (TiOx NPs) represent a variant of titanium dioxide nanoparticles (TiO_2_ NPs), produced through the direct interaction of TiO_2_ NPs with H_2_O_2_ [171]. TiOx NPs have been introduced as secure radiosensitizing agents, demonstrating proficiency in suppressing the proliferation of pancreatic cancer stem cells (CSCs) and impeding their ability to migrate and infiltrate within both laboratory and living organism environments. The recognized mechanism of action involves the generation of ROS, which deactivate the AKT signaling pathway. This pathway is responsible for the activation of transcription factors associated with stemness. Furthermore, PAA-TiOx NPs exhibit significant potential as radiosensitizing agents, enhancing the effectiveness of X-ray treatment when directly administered to tumor sites [172,173,174]. This research utilized a clonogenic assay to examine the impact of PAA-TiOx nanoparticles (NPs) and free H_2_O_2_ molecules on the radiosensitivity of cancer cells within a controlled laboratory environment. The investigation employed a cell-free dialysis technique, revealing that a fraction of the H_2_O_2_ molecules adhered to the PAA-TiOx NPs during their synthesis phase, which was subsequently released during the treatment process. A correlation was observed between the intensity of X-ray exposure and the concentration of H_2_O_2_ within the cells treated with PAA-TiOx NPs. Additionally, it was discovered that both TiOx NPs and PAA-TiOx NPs possessed a distinct ability to produce hydroxyl radicals in response to X-ray irradiation. The introduction of PAA-TiOx NPs resulted in an enhancement in DNA damage and an increase in cytotoxicity following X-ray irradiation in both in vitro and in vivo models utilizing human pancreatic cancer cells transplanted into mice, which are shown in Figure 6a. It is also noteworthy that zinc peroxides have demonstrated potential applications in cancer therapy. Furthermore, it has been discovered that zinc peroxide–iron nanocomposites, when modified with hyaluronic acid, significantly improve the effectiveness of cancer immunotherapy. This enhancement is achieved by boosting the antitumor immune response to immune checkpoint inhibitors, particularly αPD-1, which is shown in Figure 6b [175]. The utilization of nanocomposite materials has been found to contribute to the deterioration of the tumor extracellular matrix (ECM) by the generation of highly oxidative hydroxyl radicals. This phenomenon leads to a significant alteration in the structure of the tumor stromal microenvironment. As a result, this procedure enhances blood circulation, improves the delivery of therapeutic agents, and augments the penetration of immune cells into tumor cells. This improvement is accomplished by triggering pyroptosis, a type of cell death, via the activation of the caspase-1/GSDMD-dependent pathway. Moreover, the nanocomposite materials promote ferroptosis through several mechanisms, such as increasing the concentration of Fe^2+^ within the intracellular iron pool, diminishing the expression of FPN1 to hinder iron export, and activating the p53 signaling pathway to interrupt the SLC7A11-GSH-GPX4 signaling pathway. Additionally, nanoparticles enriched with manganese and zinc peroxide have been demonstrated to exhibit synergistic effects in cancer immunotherapy by promoting the immunogenic death (ICD) of cancer cells [176]. Mn^2+^-based composites have been found to activate the STING pathway in a manner that synergistically enhances the secretion of type I interferon and inflammatory cytokines, thereby facilitating specific T-cell responses. This process systematically suppresses the TME by reducing the number of regulatory T cells (Tregs) and polarizing macrophages towards an M1 phenotype, thereby initiating a cascade of adaptive immune responses. In conjunction with the anti-PD-1 antibody, the nanocomposite has been shown to exhibit an enhanced capacity for tumor growth suppression and the prevention of lung metastasis. The research team, under the leadership of Wang, has pioneered a groundbreaking carrier-free nanoparticle system named Mn-ZnO_2_. This innovative system employs a synergistic strategy in cancer therapy by simultaneously releasing two ions, Mn^2+^ and Zn^2+^, along with ROS, thereby influencing the degradation and activation of the p53 protein at the site of the tumor. The acidic cellular milieu facilitates the swift release of H_2_O_2_ and Zn^2+^ by these nanoparticles, resulting in an enhancement of endogenous ROS production and the disruption of the mutant p53 protein (Mutp53), in addition to the formation of detrimental hydroxyl radicals (•OH), through a Fenton-like reaction. This nanoparticle, devoid of the need for a carrier, stands as a simple and efficacious approach toward the creation of multifunctional nanosystems [177].

**Table 1 cancers-16-03581-t001:** Summary of various metal peroxide nanoparticles and their applications for cancer therapy.

MO_2_	Surface Modifier	Application	Targeting Approach Passive or Active	Model (In Vivo and In Vitro)	Ref
CuO_2_	PVP	CDT	Passive	Both	[132]
CuO_2_	PLGA	PTT	Passive	Both	[133]
CuO_2_	luminol	CDT/PDT	active	Both	[134]
CuO_2_		CDT	Passive	In vitro	[135]
CuO_2_	PVP	CDT/ Starvation therapy	active	In vitro	[136]
CuO_2_	PVP	CDT	Passive	In vitro	[137]
CuO_2_		CDT	Passive	In vitro	[138]
CuO_2_	CMC	CDT/PDT	Passive	Both	[139]
CuO_2_	CCM	CDT/PDT/PTT	Passive	Both	[140]
CuO_2_	SiO_2_	CDT/cuproptosis/ chemotherapy	Passive	Both	[142]
CuO_2_	5-poly(amidoamine) dendrimer	ferroptosis/cuproptosis/CDT	Active	Both	[142]
CaO_2_	ferrocene	calcium overload/CDT	passive	Both	[143]
CaO_2_	Hyaluronic acid	Calcification	Active	Both	[144]
CaO_2_	*ε*-Poly Lysine and Hyaluronic Acid	PDT/calcium overload	Active	Both	[145]
CaO_2_	PCM	calcium overload/calcification	Passive	Both	[146]
CaO_2_	Sodium alginate	PTT/calcium overload	Passive	Both	[147]
CaO_2_	liposomes	PDT	Passive	Both	[148]
CaO_2_	sodium-hyaluronate	Calcium overload	Passive	Both	[150]
CaO_2_	ZIF-67	Chemo/CDT	Passive	Both	[153]
CaO_2_	Lauric acid	PDT/CDT	Passive	Both	[154]
CaO_2_	solid lipid monostearin	Chemo/CDT	Active	Both	[155]
CaO_2_	Hyaluronic acid	calcicoptosis therapy/chemotherapy	Active	Both	[156]
CaO_2_	phospholipid-coated liposomes	PDT/PTT	Passive	Both	[157]
TiOx		radiotherapy	Passive	Both	[158]
TiOx	PAA	Radiotherapy	Passive	In vitro	[159]
TiOx	PAA	Radiotherapy	Passive	Both	[160]
TiOx	PAA	Radiotherapy	Passive	Both	[161]
Fe-ZnO_2_	Hyaluronic acid	Ferroptosis/pyroptosis Immunotherapy	Passive		[162]
Mn-ZnO_2_		Immunotherapy	Passive		[163]
Mn-ZnO_2_		dual ions and ROS	Passive	Both	[164]

## 5. Biosafety

The biological effects and biosafety considerations associated with metal peroxide nanoparticles play a pivotal role in assessing their potential for clinical application. Although these nanoparticles have shown considerable promise in terms of therapeutic efficacy within the realm of chemoreactive nanotherapeutics, their reactivity and elemental composition could introduce potential risks of toxicity and adverse effects. For example, the high reactivity of metal peroxides may catalyze reactions with the surrounding water molecules in the bloodstream, leading to the generation of H_2_O_2_. This process could potentially harm healthy cells and tissues, thereby necessitating a thorough evaluation of their safety profile [178,179]. Moreover, the creation of metal peroxides and the ongoing discharge of metal ions may also play a role in raising toxicity concerns. This is because specific metal-ion species, such as Cu^2+^, Zn^2+^, and Ba^2+^, are recognized to possess toxic properties and have the potential to damage healthy tissues [180,181,182,183].

Two strategies have been suggested to mitigate potential toxicity and adverse side effects, thereby facilitating further foundational research. To control the undesirable reactivity, it is advisable to employ suitable surface modifications and encapsulation of nanocarriers to modulate the reactivity of metal peroxides as shown in Table 1. This strategy is designed to confine chemical reactions to the vicinity of the disease microenvironment, thereby sparing blood vessels and healthy tissues. Moreover, to avert the possible release of toxic metal ions, it is critical to ensure the controlled decomposition of metal peroxide nanoparticles. Specifically, the decomposition process should occur under mildly acidic conditions within tumor environments, as opposed to in normal, neutral tissues. Even after the decomposition of metal ions from the metal peroxides, the biological actions of metal ions were irregular. In particular, irregular distribution impacts the different functions of the cells, resulting in negative consequences and also death [184]. A further restriction is the possibility that metal ions may build up in the body gradually, especially in organs such as the liver and spleen, which could lead to lasting health consequences. Other challenges in applying metal ions for therapy involve a prolonged process for developing drugs, poor efficiency in loading drugs into NPs, limited absorption by cells, and the inability to convert findings from lab experiments to real-world trials [185,186]. It has been verified that the majority of cancer cells are susceptible to metal ions, rendering them a suitable choice for multiple-course therapy. Moreover, metal ions have expanded their use in fighting cancer with the progress of nanotechnology [127]. Although numerous metal ions can cause the death of tumor cells by producing ROS, it appears that the low-oxygen and highly reducing environment of the TME is not favorable for this process. It is important to consider how to maintain the right pH and sufficient levels of H_2_O_2_ to initiate the catalytic pathway. Additionally, it is crucial to meticulously consider and tailor the commonly accepted targeting method to guarantee that the significant accumulation of metal peroxide nanoparticles at lesion sites effectively diminishes their impact and adverse effects on healthy cells and tissues.

The preliminary examination of the biocompatibility and biosafety of metal peroxide nanoparticles has been conducted through various methodologies. Although the initial results are encouraging, they fall short of providing a conclusive demonstration of biosafety, which is essential for the progression of clinical applications. It is, therefore, imperative to conduct more extensive in vitro and in vivo biosafety evaluations to generate robust data and evidence concerning biocompatibility and biosafety [147,187,188,189].

Metal peroxides with various properties, including high reactivity and difficulties in fabrication, result in irregular morphology, uncontrollable particle size, and easy aggregation, which will be challenging for their stability. In addition, reactions with water always result in difficulty in storage because they can slowly react with surrounding water molecules, resulting in uncontrollable nanoparticle quality for further biomedical use. The reactivity is also doubtful due to their initial stage of development, which affects healthy cells and is uncontrollable. Biocompatibility should also be taken into account when synthesizing organic and inorganic peroxide nanoparticles. Metal peroxides are in the initial stage; they cannot be applicable to a wide range of biomedical applications, which can be reduced or nullified as time progresses in personalized nanomedicine.

Future systematic fundamental research is anticipated to elucidate aspects such as in vivo biodistribution, excretion, and histocompatibility, particularly long-term biological effects.

## 6. Conclusions and Outlook

Despite the presence of numerous unresolved challenges, metal peroxides have emerged as pioneers in the development of innovative strategies for tumor treatment. This progress underscores the need for further exploration and development within the biological sciences. Metal peroxide nanostructures have been successfully synthesized and utilized as sources of oxygen and H_2_O_2_ within the cancer TME, yielding promising outcomes. In acidic conditions, the generation of H_2_O_2_, which occurs through the interaction of metal peroxide with water, serves two distinct purposes. Initially, it induces oxidative stress, and subsequently, it produces a surplus of oxygen via reactions with molecules such as catalase enzymes. This method helps in reducing the effects of low oxygen in tumors, thus tackling the common issue of diminished oxygen found in the area surrounding the tumor [190,191,192]. Furthermore, the characteristics of metal peroxides can be efficiently incorporated into a diverse array of materials, including photosensitizers, enzymes, metal nanoparticles, Fenton reagents, and chemotherapeutic drugs. This incorporation facilitates the creation of combination therapies, encompassing photodynamic therapy, CDT, and chemotherapy [193,194,195]. The co-administration of metal peroxide-based treatments with other therapeutic modalities has demonstrated significantly enhanced antiproliferative effects. Understanding the potential response mechanisms of metal ion-based agents at the specific tissue, single cell, and molecular levels can only help researchers study the properties and fate of these ions. In addition, multifunctional optical probes have been developed over the years for the detection of the concentration, distribution, and interaction of ions with specific biochemical substances in the biological environment. It may help to accurately pinpoint the metabolic changes induced by metal ions in vivo, which can be applicable in the biological system for improved therapeutic efficiency against various diseases.

While not exhaustive, this review highlights the most extensively studied metal peroxide nanosystems utilized in the domain of cancer research. It provides a thorough examination of their applications within the cellular milieu of noncancerous tumors.

## Figures and Tables

**Figure 1 cancers-16-03581-f001:**
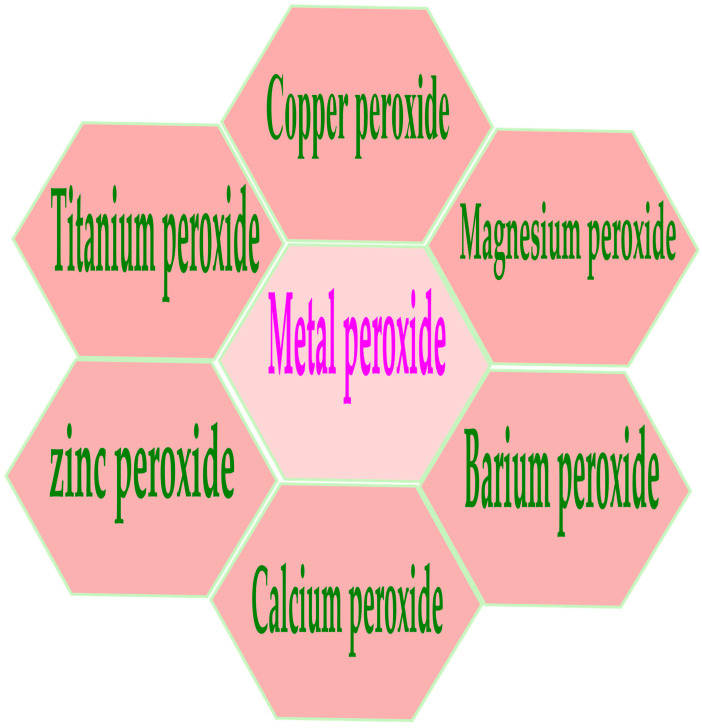
Illustrates various metal peroxides, including copper peroxide (CuO_2_), calcium peroxide (CaO_2_), magnesium peroxide (MgO_2_), zinc peroxide (ZnO_2_), barium peroxide (BaO_2_), and titanium peroxide (TiOx).

**Figure 2 cancers-16-03581-f002:**
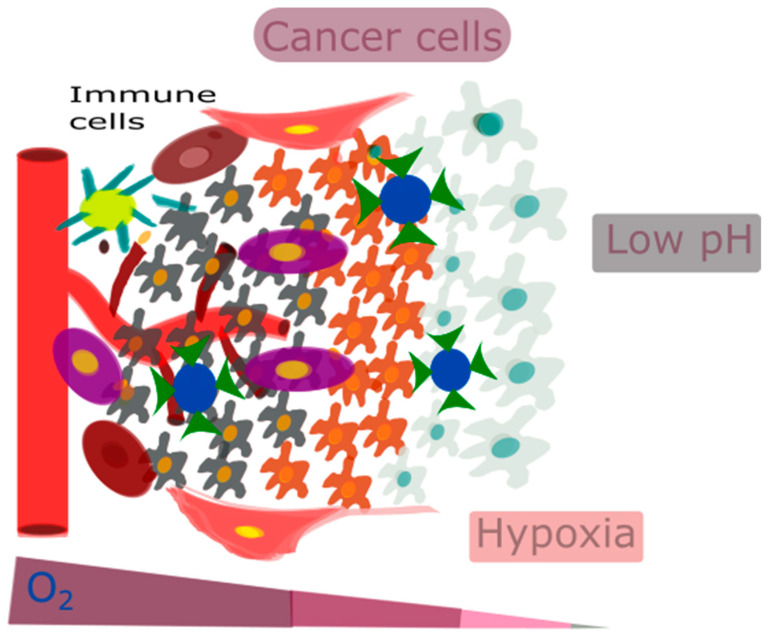
The characteristic features of the TME, such as low pH, hypoxia, and an immunosuppressive environment.

**Figure 3 cancers-16-03581-f003:**
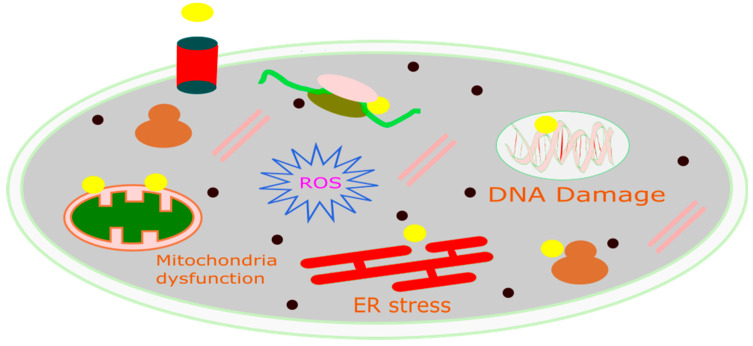
Schematic diagram of the mechanism of CDT for cancer treatment.

**Figure 4 cancers-16-03581-f004:**
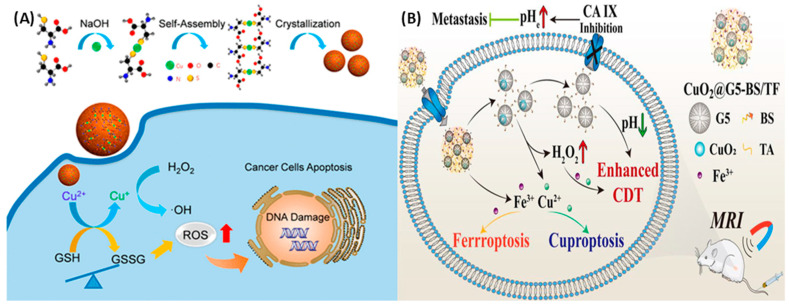
(**A**) Schematic illustration of the preparation of self-assembled copper–amino acid mercaptide nanoparticles (Cu-Cys NPs). In situ glutathione-activated and H_2_O_2_-reinforced CDTs were achieved using the Cu-Cys NPs with the help of the Fenton reaction +. Reprinted with permission from Ref. [138]. 2019, AMERICAN CHEMICAL SOCIETY. (**B**) Schematic illustration of CuO_2_@G5-BS/TF nanocomplexes against the TME. The nanocomplex achieves cell death by Enhanced CDT, Ferroptosis, and Cuprotosis due to the presence of Fe^3+^ and Cu^2+^. Reprinted with permission from Ref. [142]. 2024.

**Figure 5 cancers-16-03581-f005:**
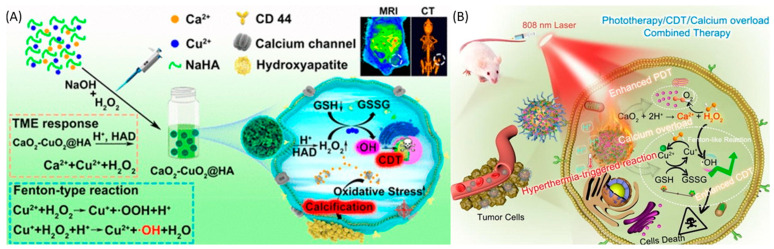
(**A**) Schematic illustration of a hyaluronic acid-modified calcium and copper peroxide nanocomposite against the TME. Enhanced tumor therapy was achieved through the synergistic effect of a Fenton-like reaction by Cu^2+^ and mitochondria dysfunction by Ca^2+^ in ROS generation through the nanocomposite. Reprinted with permission from Ref. [144]. 2022, AMERICAN CHEMICAL SOCIETY”. (**B**) Schematic illustration of a CaO_2_-Cu/ICG@PCM nanoplatform against the TME. The nanocomplex achieves bioimaging and tumor therapy through a combinational treatment of PDT, CDT, and calcium overload, along with CT imaging. Reprinted with permission from Ref. [154]. 2021, ELSEVIER, ELSEVIER.

**Figure 6 cancers-16-03581-f006:**
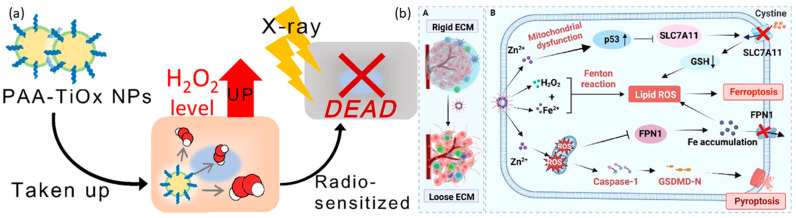
(**a**). Schematic illustration of PAA–TiOx nanocomplexes against tumors. After take-up by the cancer cells, the nanocomplex achieves cell death by radiotherapy. “Reprinted with permission from Ref. [159]. 2021, ELSEVIER” (**b**). A. Schematic illustration of Fe-ZnO_2_@HA nanoparticles shows the changes in the rigid extracellular membrane after Fe-ZnO_2_@HA enters into the tumor environment. B. Fe-ZnO_2_@HA represents the mechanism of action of Fe-ZnO_2_@HA to induce apoptosis through Ferroptosis and Pyroptosis. “Reprinted with permission from Ref. [162]. 2024, AMERICAN CHEMICAL SOCIETY”.

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
