# Peer review of "Metal Peroxide Nanoparticles for Modulating the Tumor Microenvironment: Current Status and Recent Prospects"

_cancers, 2024, doi:10.3390/cancers16213581_

Round 1
Reviewer 1 Report
Comments and Suggestions for Authors
The review manuscript has documented the advancements in the utilization of metal peroxide nanoparticles for improvement of cancer therapeutics. This review is definitely very timely and important. However, there are few concerns which needs to be addressed before assessing the suitability of the manuscript for publication. My specific comments are appended below.
1. Based on the title of the article, the authors are encouraged to present their perspective as well on the state of art and future outcomes, in depth, which will contribute to a further comprehensive understanding of the readers regarding the topic.
2. The major challenges of metal peroxide nanoparticles for clinical translation have not been adequately discussed. Stability and toxicity concerns and regulatory landscape should also be explored extensively.
3. The figures look slightly distorted (for example, figure 1, 2 and 3). Please ensure to provide high resolution quality figures in the manuscript.
Comments on the Quality of English LanguageThe quality of the English language needs some improvement. It has to be polished and simplified.
Author Response
Reviewer 1:
We appreciate the efforts of the reviewers for their detailed and insightful comments, which have helped us improve the quality of our manuscript. We have appended a detailed response to the reviewer's comments below for your convenience.
The review manuscript has documented the advancements in the utilization of metal peroxide nanoparticles for improvement of cancer therapeutics. This review is definitely very timely and important. However, there are few concerns which needs to be addressed before assessing the suitability of the manuscript for publication. My specific comments are appended below.
Q.1 Based on the title of the article, the authors are encouraged to present their perspective as well on the state of art and future outcomes, in depth, which will contribute to a further comprehensive understanding of the readers regarding the topic.
Answer: We agree with the reviewer’s suggestion. The future outcomes for metal peroxides-based nanoparticles were explained in the revised manuscript for the better understanding for the readers in terms of various components of TME and their role in development of cancer in lines 184-190,194-201, 236-247 in the revised manuscript.
Q.2 The major challenges of metal peroxide nanoparticles for clinical translation have not been adequately discussed. Stability and toxicity concerns and regulatory landscape should also be explored extensively.
Answer: The metal oxide nanoparticles biosafety and their clinical translation were described in order to provide a better view towards the biomedical application of metal peroxides. From lines 894-908, 920-929, 951-958 of the revised manuscript.
Q.3 The figures look slightly distorted (for example, figure 1, 2 and 3). Please ensure to provide high resolution quality figures in the manuscript.
Answer: Thanks for pointing out the mistake. The figures 1, 2, 3 used in the article were changed with high resolution images.
Q.4 Comments on the Quality of English Language. The quality of the English language needs some improvement. It has to be polished and simplified
Answer: A native English speaker carefully checked our revised manuscript for grammatical and typographic errors. Thank you for the suggestion.
Reviewer 2 Report
Comments and Suggestions for Authors
Reviewer Comments to Author(s)
Recommendation: Major revisions
In this review the authors present the application of metal oxide nanoparticles for targeting the tumor microenvironment. The authors present a great variety of research works on metal oxides but they are advised to be more careful and present the scientific background of TME in more detail and coherence. The following comments will help the authors.
The title of the article is referring to cancer therapy; however, the main part is targeting of the TME. Maybe the authors should consider a rephrasing of the title that better describes the aim of the review article.
Moreover, extensive corrections should be provided throughout the article in the correct use of subscript or superscript. All the metal peroxide chemical formulas should be with a subscript in the oxygen and hydrogen peroxide. All the ions should be with a superscript for the ionic number. Please provide the correct forms of the radicals, as well. Provide such corrections throughout the review article.
In the abstract the authors use the phrase “to the initiation of chemical reactions”. The meaning is a little weird to follow (it is mentioned in line 10 and 19). Could you please either rephrase or correct depending on the meaning that should be provided? Moreover, have the authors consider providing a less detailed and more coherent abstract?
Please provide syntactic and grammatical corrections throughout the article. The authors should be careful with the use of words and terms. For example, line 38 the scientific term used id DNA abnormalities or DNA damage (as mentioned by the authors in line 41), when such mistakes are in the first line they might probably create a negative sense to the reader. Another example, line 41-42 “Researchers identify DNA damage as the foremost element that…”. The use of element in the sentence is not correct, since in DNA sequences there are chemical elements. Probably the authors mean the foremost factor/sign/cause/reason. The authors are kindly asked to provide such corrections throughout the article, when necessary.
In lines 53-54 the authors say that chemo drugs may unintentionally cause adverse effects. However, the use of “may” is not correct since it is well-known that chemo drugs do cause and in some cases fatal severe adverse effects. Moreover, the reason is the lack of specificity and targeting ability. Since the authors do present targeting metal oxide nanoparticles for TME, they could refer to the reason behind the adverse effects and the promotion of nanotechnology as an answer.
In lines 57-58, surgical ablation (better than surgical therapy) is associated with the drawback of the remaining cancer cells that increase the possibility of cancer resurfacing, progression and metastasis. The specialized knowledge can not be considered as a drawback. Metastasis that is highly linked with TME is only mentioned much later in part 2.1. The authors are advised in the Introduction to emphasize in the aim of their review article through relations, as conventional therapies – nanotechnology, TME – metastatic potential, TME – nanotechnology, TME – metal oxides.
The authors repeatedly mention the abbreviation “tumor microenvironment (TME)”. The abbreviations are explained the first time the term is presented and used thereafter. Please correct accordingly. (lines 124, 129, 130, 140, 150 and throughout the article). Such corrections should be followed and for other terms, including hydrogen peroxide (H2O2), chemodynamic therapy (CDT), reactive oxygen species (ROS) and more.
Figure 1 is not clear
In part 2, the authors present the TME and refer to cellular and non-cellular components of the TME and their interactions without specifying any of these concepts. Later, they refer to endothelial cells, fibroblasts, immune cells (types?) and immune responses. Which is the connections of these types of cells with the TME and which are the types of interactions they affect/interfere or regulate? Are all these types of cells cancerous? Are there non-cancerous cells or stem cells?
In lines 132-134: How the immune cells have both adaptive and innate components? Are there types of immune cells that are linked to the adaptive and innate immune system? And how the adaptive and innate immune system regulates the therapeutic responses?
In line 176 – 187: the HIFs are a family of transcriptional factors composed of more than HIF-1alpha that regulate the responses of cancer cells in the hypoxic environment of TME. HIFs’ are related to cancer cell survival, growth, gene expression and even cancer pathobiology, including abnormal angiogenesis, metabolism, proliferation, and invasion. Why the authors have chosen to be so short in this field that is highly related to the effect of metal oxide nanoparticles through CDT and fenton reaction? Moreover, in the part that the Fenton reaction is presented it is not clear how this reaction is connected with the TME, since Fenton reaction and the levels of ferrous ions are linked to ROS levels, ferroptosis and thought HIF family to the regulation of tumor genes expression.
Concerning the biosafety of metal oxide nanoparticles, a great implication in their therapeutic application is in connection with metabolic pathways cancer metabolism, since metal ions are considered great regulators of cellular metabolic pathways. Probably this concern is greater than their toxicity, since the surface modification will probably enhance the localization of the metal oxides at TME with low toxicity. However, after decomposition of the surface which is the fate of the metal oxides and how the increased concentration of metal ions will affect cell metabolism?
Comments on the Quality of English LanguageThe authors are advised to correct the research/scientific terms used in some cases. the comments provide detailed description of the suggested corrections.
Author Response
We appreciate the efforts of the reviewers for their detailed and insightful comments, which have helped us improve the quality of our manuscript. We have appended a detailed response to the reviewer's comments below for your convenience.
In this review the authors present the application of metal oxide nanoparticles for targeting the tumor microenvironment. The authors present a great variety of research works on metal oxides but they are advised to be more careful and present the scientific background of TME in more detail and coherence. The following comments will help the authors.
Q.1. The title of the article is referring to cancer therapy; however, the main part is targeting of the TME. Maybe the authors should consider a rephrasing of the title that better describes the aim of the review article.
Answer: We appreciate this valuable suggestion from the reviewer. The title is revised to “Metal peroxide nanoparticles for the modulation of tumor microenvironment: current status and recent prospects”
Q.2. Moreover, extensive corrections should be provided throughout the article in the correct use of subscript or superscript. All the metal peroxide chemical formulas should be with a subscript in the oxygen and hydrogen peroxide. All the ions should be with a superscript for the ionic number. Please provide the correct forms of the radicals, as well. Provide such corrections throughout the review article.
Answer: Thank you for the advice. We fixed the chemical formulas for all the metal peroxides in the revised manuscript with subscripts for oxygen and hydrogen peroxide and also all ions have superscript for the ionic number. We also made sure that the radicals were in the right form by using scientific language.
Q.3. In the abstract the authors use the phrase “to the initiation of chemical reactions”. The meaning is a little weird to follow (it is mentioned in line 10 and 19). Could you please either rephrase or correct depending on the meaning that should be provided? Moreover, have the authors consider providing a less detailed and more coherent abstract?
Answer: Thank you for the suggestion. We corrected the mistake by using the correct phrase in the abstract, which will help in understanding the meaning of the sentences easier. We also changed the abstract to maintain coherence. We highlighted all the changes in the red font color for better understanding and shortened the abstract. Lines 9-34.
Q.4. Please provide syntactic and grammatical corrections throughout the article. The authors should be careful with the use of words and terms. For example, line 38 the scientific term used id DNA abnormalities or DNA damage (as mentioned by the authors in line 41), when such mistakes are in the first line, they might probably create a negative sense to the reader. Another example, line 41-42 “Researchers identify DNA damage as the foremost element that…”. The use of element in the sentence is not correct, since in DNA sequences there are chemical elements. Probably the authors mean the foremost factor/sign/cause/reason. The authors are kindly asked to provide such corrections throughout the article, when necessary.
Answer: Thank you for pointing out the mistakes. We have asked a native English speaker to thoroughly check the revised manuscript to correct all the grammar mistakes in it. The revised manuscript now includes corrections to lines 41-42.
Q.5. In lines 53-54 the authors say that chemo drugs may unintentionally cause adverse effects. However, the use of “may” is not correct since it is well-known that chemo drugs do cause and, in some cases, fatal severe adverse effects. Moreover, the reason is the lack of specificity and targeting ability. Since the authors do present targeting metal oxide nanoparticles for TME, they could refer to the reason behind the adverse effects and the promotion of nanotechnology as an answer.
Answer: Thank you for the suggestion. We have rephrased the sentence in lines 53-54 to “Chemotherapeutic drugs unintentionally cause adverse and fatal side effects when they target healthy tissues such as the blood and digestive tract." Lines 53-54 of the revised manuscript reflect the revision. Additionally, we have mentioned the limitations of conventional cancer treatments in lines 54-62 and the advantages of nanomedicine in cancer in lines 81-83.
Q.6. In lines 57-58, surgical ablation (better than surgical therapy) is associated with the drawback of the remaining cancer cells that increase the possibility of cancer resurfacing, progression and metastasis. The specialized knowledge cannot be considered as a drawback. Metastasis that is highly linked with TME is only mentioned much later in part 2.1. The authors are advised in the Introduction to emphasize in the aim of their review article through relations, as conventional therapies – nanotechnology, TME – metastatic potential, TME – nanotechnology, TME – metal oxides.
Answer: We thank the reviewer for this valuable comment. We have revised the title of the manuscript to “Metal peroxide nanoparticles for the modulation of tumor microenvironment: current status and recent prospects”. In the revised manuscript introduction section, we have emphasized the challenges of conventional therapy from lines 53-63, TME – nanotechnology from lines 88-101 and TME – metal oxides from lines 111-117 in the revised manuscript.
Q.7. The authors repeatedly mention the abbreviation “tumor microenvironment (TME)”. The abbreviations are explained the first time the term is presented and used thereafter. Please correct accordingly. (lines 124, 129, 130, 140, 150 and throughout the article). Such corrections should be followed and for other terms, including hydrogen peroxide (H2O2), chemodynamic therapy (CDT), reactive oxygen species (ROS) and more.
Answer: The abbreviations full form was mentioned only for the first time in the revised manuscript for TME, CDT and ROS. The metal oxides are correctly represented in the revised manuscript. Thank you for the suggestion.
Q.8. Figure 1 is not clear
Answer: Thanks for pointing it out. Clear image was provided with correct figure legend.
Q.9. In part 2, the authors present the TME and refer to cellular and non-cellular components of the TME and their interactions without specifying any of these concepts. Later, they refer to endothelial cells, fibroblasts, immune cells (types?) and immune responses. Which are the connections of these types of cells with the TME and which are the types of interactions they affect/interfere or regulate? Are all these types of cells cancerous? Are there non-cancerous cells or stem cells?
Answer: Cellular and non-cellular components TME were explained in the revised manuscript from lines 161-174. The TME components were explained in the article were based on the cancer cells and their role in the progress of cancer.
Q.10. In lines 132-134: How the immune cells have both adaptive and innate components? Are there types of immune cells that are linked to the adaptive and innate immune system? And how the adaptive and innate immune system regulates the therapeutic responses?
Answer: The immune cells have two types of response such as innate and adaptive response in order to fight against the pathogens which enter into the body and leads to various diseases. Fighting against the abnormalities required by the immune system in order to regulates the therapeutic responses. The original sentence is revised in lines 162-165 of the revised manuscript with a proper citation (Citation 42) about innate and adaptive immune responses.
Q.11. In line 176 – 187: the HIFs are a family of transcriptional factors composed of more than HIF-1alpha that regulate the responses of cancer cells in the hypoxic environment of TME. HIFs’ are related to cancer cell survival, growth, gene expression and even cancer pathobiology, including abnormal angiogenesis, metabolism, proliferation, and invasion. Why the authors have chosen to be so short in this field that is highly related to the effect of metal oxide nanoparticles through CDT and fenton reaction? Moreover, in the part that the Fenton reaction is presented it is not clear how this reaction is connected with the TME, since Fenton reaction and the levels of ferrous ions are linked to ROS levels, ferroptosis and thought HIF family to the regulation of tumor genes expression.
Answer: We have explained the types of HIFs in the HIFs family response for the cancer cell survival, growth, gene expression and even cancer pathobiology in the line 236-240. Also, the metal oxide nanoparticles response to TME were also well described in the revised manuscript from the line 375-384
Q.12. Concerning the biosafety of metal oxide nanoparticles, a great implication in their therapeutic application is in connection with metabolic pathways cancer metabolism, since metal ions are considered great regulators of cellular metabolic pathways. Probably this concern is greater than their toxicity, since the surface modification will probably enhance the localization of the metal oxides at TME with low toxicity. However, after decomposition of the surface which is the fate of the metal oxides and how the increased concentration of metal ions will affect cell metabolism?
Answer: The metal oxide nanoparticles biosafety and their clinical translation were described in order to provide a better view towards the biomedical application of metal peroxides. From lines 894-908, 920-929, 951-958.
Round 2
Reviewer 1 Report
Comments and Suggestions for Authors
The authors have addressed most of the concerns raised by the reviewers. However, the figures seems to be still stretched. This can be corrected before recommending the manuscript for publication.
Reviewer 2 Report
Comments and Suggestions for Authors
All the comments and suggestions were followed by the authors.